# Overall Design of Satellite Networks for Internet Services with QoS Support

**Kyu-Hwan Lee** and **Kyoung Youl Park** *

Agency for Defense Development, DaeJeon 43186, Korea; drkhlee@add.re.kr
* Correspondence: kypark@add.re.kr

**Abstract:** The satellite network is useful in various applications because of its coverage, broadcast capability, costs independent of the distance, and easy deployment. Recently, thanks to the advanced technologies in the satellite communication, the high throughput satellite system with mesh connections has been applied to the Internet backbone. In this paper, we propose a practical overall design of the satellite network to provide Internet services with quality of service (QoS) support via the satellite network. In the proposed design, we consider two service types such as delay-tolerance and delay-sensitive services allowing the long propagation delay of the satellite link. Since it is crucial to evaluate the user satisfaction in the application layer for various environments to provide the QoS support, we also define the performance metrics for the user satisfaction and derive the major factors to be considered in the QoS policy. The results of the performance evaluation show that there are factors such as the traffic load and burstiness in the QoS policy for the delay-tolerance service with volume-based dynamic capacity in the satellite network. For the delay-sensitive service with rate-based dynamic capacity, it is additionally indicated that it is important for the estimation of effective data transmission rate to guarantee the QoS. Furthermore, it is shown that the small data size has an effect on the slight reduction of the QoS performance in the satellite network.

**Keywords:** satellite network; MF-TDMA; mesh connection; QoS; user satisfaction

---

## 1. Introduction

In the future network, it will be necessary to offer tremendously more capacity than current ones to satisfy increasing traffic demanded by users and new applications with the quality of service (QoS) [1–4]. Especially, network operators will face a challenge in providing Internet service for users in rural or other difficult-to-serve areas while guaranteeing the QoS [1]. A possible solution is a satellite network to address this challenge. Broadband satellite networks have come to play a major role in overall communication because of their wide coverage area and reliable links [5–8]. The satellite network is a cost-effective solution for providing communication services to other difficult-to-serve areas. In recent years, thanks to the advanced technologies such as on-board processing, the dynamic resource allocation, the spot beam technology, and the use of a higher radio frequency, high throughput satellite (HTS) systems using the geostationary orbit (GEO) satellites with mesh connections can be used for the Internet backbone [5–8]. Currently, the ViaSat-2 service using the GEO satellite has provided Internet services with the capacity of 300 Gbps [9]. Next-generation HTS systems are targeting terabit/s aggregated capacity. This system is expected to meet increasing demand of Internet services for the Internet of Things (IoT), such as remote control, telemetry, and surveillance of sensing data [10]. Furthermore, it is useful in supporting multimedia, unmanned aircraft systems and urgent data services [6,8].

In the satellite network to support the QoS, inherent characteristics such as a long propagation delay and use of a higher frequency should be addressed. Substantial research has focused on the

needed technologies to provide Internet services satisfying users in the satellite network [10–22]. In [11], a resource allocation scheme including a packing algorithm was proposed to provide QoS connections over multi-frequency time division multiple access (MF-TDMA) satellite systems. The authors of [12] considered a mesh satellite networking with turbo coding, on-the-move (OTM) terminal, and class-based weighted fair queuing with Internet protocol (IP) QoS. The authors of [13] suggested a code design for the mobile digital video broadcasting-return channel via satellite (DVB-RCS) and evaluated the code design in terms of QoS performance. In [14], the power control for video communication via satellite links was addressed for considering QoS in the environment of rain attenuation. The authors of [15] considered the gateway handover and network coding for the impaired gateway in the smart gateway diversity of satellite networks for the QoS support. In [16], QoS-based admission control using a multipath scheduler was proposed for IP over satellite networks. The authors of [17] proposed the distributed QoS awareness in the satellite communication network with optimal routing to consider inter-satellite links. The authors of [18] proposed the graph-based QoS support routing strategy for satellite networks with the on-demand transmission scenario. In [19], the transmission control protocol (TCP) accelerator for DVB-RCS networks with IP encryption was addressed. In [20,21], the congestion control mechanism and the use of caches were considered in the satellite network with the QoS support. In [10,22], satellite communications supporting IoT with the QoS support were addressed. In [23,24], voice over IP (VoIP) services to guarantee the QoS were considered in the satellite network.

To provide Internet services with the QoS support via satellite networks, it is necessary to estimate the network level performance for the satellite network, which consists of the end-to-end network architecture. However, to the best of our knowledge, that whole network-level performance evaluation and design for the QoS in the satellite network according to various applications of Internet services have not been explored. To provide Internet services with the QoS support via satellite networks, the overall architecture of the satellite network should be considered practically [1,25,26]. Furthermore, it is crucial to evaluate the practical network performance considering control and protocol overheads. It is also important to evaluate the user satisfaction in the application layer for various environments to provide the QoS support. Therefore, in this paper, we consider a widely-used MF-TDMA satellite network with mesh connections [11,12,25,26]. We also design the overall satellite network to provide Internet services with the QoS support practically. In the proposed design, there are two service types such as delay-tolerant and delay-sensitive services allowing the long propagation delay of the satellite link. To evaluate the user satisfaction with services, we define the performance metrics in the application layer and measure the network performance for various environments by varying each factor such as the service type, the traffic load, the burstiness of the traffic, the number of users, and the resource allocation method in the link layer. We then derive major factors to be considered in the QoS policy. The main contributions of our paper are as follows:

1. Practical overall design of the MF-TDMA satellite network for Internet services with the QoS support.
2. Practical and meaningful performance analysis considering control and protocol overheads.
3. Performance evaluation of the user satisfaction for various environments in the application layer and derivation of major factors to be considered in the QoS policy.

The remainder of this paper is organized as follows: Section 2 presents the system model and the proposed overall network design. Section 3 describes a performance evaluation of the proposed overall network design. Finally, the conclusion is presented.

## 2. Proposed Design of Satellite Networks

To design and evaluate the whole satellite network guaranteeing the QoS, it is necessary that the design of link and network layers in the satellite network precedes [1,11,13,23]. In this section, we first address our customized design of the link and the network layers in the satellite network

based on the DVB-RCS system, which is a widely-used satellite system [11,12,25–35]. We also suggest the QoS architecture for the satellite network and discuss the other considerations to provide the QoS support in the satellite network. In this paper, we consider the satellite with on-board processing. We make the following assumptions regarding the proposed design:

1. Based on the DVB-RCS system, the satellite network in the paper consists of a GEO satellite, a hub terminal with a network control center (NCC) and a network management center (NMC) and user terminals (UTs), as shown in Figure 1 [11,12,25–27]. The terrestrial network between UTs is not considered.
2. In the link layer, the resource allocation scheme is demand assigned multiple access (DAMA) with the MF-TDMA technology based on the DVB-RCS system [11,12,25,27].
3. In the satellite, the uplink data channel (UDC) is switched to the downlink data channel (DDC). Thus, the mesh connection is provided to satellite terminals, as shown in Figure 1.
4. For the network control, there are a forward control channel (FCC), a return control channel (RCC), and a log-on channel (LOC). The FCC is used to broadcast control messages from the hub terminal with the NCC and the NMS to UTs. The RCC and the LOC are used to transmit control messages from UTs to the hub terminal with the NCC and the NMS. For the synchronization of the MF-TDMA in all UTs and hub terminals, a ranging channel (RC) is used to correct time and frequency synchronization.

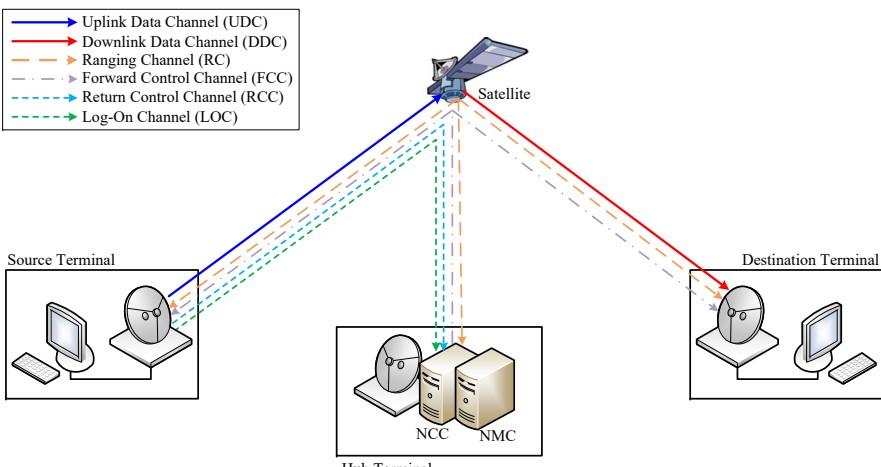

**Figure 1.** System model of the proposed design (link layer). NCC, network control center; NMC, network management center.

The detailed design of the link and the network layers is explained in the following sections.

### 2.1. Design of Link Layer

In the satellite network, the frequency resource should be efficiently used due to the limited resources. Thus, the fixed frequency band is not allocated to all UTs [11,12,25]. In the proposed design, all UTs initially execute the log-on process, and the DAMA with the MF-TDMA is used for the resource allocation to transfer data [29,31]. In the log-on process, the UT sends the log-on request message to the NMS via the LOC. The LOC is the common channel shared by the slotted aloha scheme. If NMS admits the UT into the satellite network, the needed information such as the assigned RCC and the QoS policy is transmitted to the UT via the FCC. In the NCC, the information on the UT is registered. The FCC is the dedicated channel to the hub terminal with the NCC and the NMS. After the log-on process, the UT can request the resource allocation to the NCC via the RCC. The RCC is a dedicated channel to each UT. The information such as the status of the UT and the failure of the UT is also reported to the NCC and the NMS periodically. Based on this information, the NCC allocates the resource

by each channel environment of the UT, and the NMS monitors and manages UTs. In the resource allocation for the efficient frequency usage and the QoS support, a continuous resource assignment (CRA), a rate-based dynamic capacity (RBDC), and a volume-based dynamic capacity (VBDC) are used [28–31]. In the VBDC, there are resource allocation delay including the propagation delay in the satellite link, the transmission of resource allocation request/response, the processing time for resource allocation in the NCC, and the generation of the MF-TDMA frame in UTs [28–31]. The NCC calculates the resource allocation from request messages of UTs during the super-frame and transmits the response message to each UT in the next super-frame to use resources efficiently. Thus, in the proposed design, the delay in the resource allocation of the VBDC needs to be from three to four super-frames. The information on the resource allocation for each super-frame in the MF-TDMA is broadcast to all user terminals by a terminal burst time plan (TBTP) in the FCC.

*2.2. Design of the Network Layer*

The proposed design of the network layer is shown in Figure 2. In the proposed design for the routing, the hierarchical structure is considered as the network architecture. The satellite network can be divided into the sub-autonomous system (AS) by each service provider or each beam. If all UTs share routing information among each other, a large resource is consumed to share it [32–35]. Thus, in the same AS, an open shortest path first (OSPF) routing scheme is used, and a border gateway protocol (BGP) is used between ASs for the routing, as shown in Figure 2 [34,35]. In the proposed design, for source and destination terminals in the same AS, there is a one-hop delay in the data transmission between source and destination terminals. On the other hand, when source and destination terminals are located in different ASs, respectively, multi-hop delay is needed for the data transmission via hub terminals.

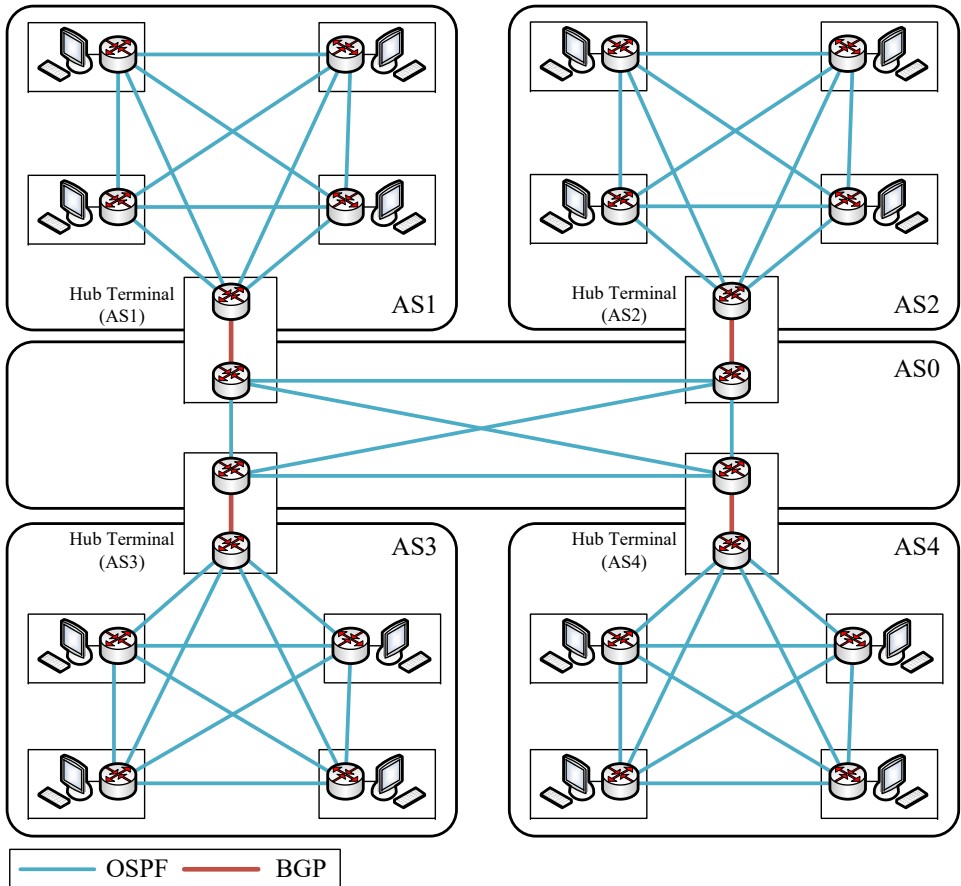

**Figure 2.** System model of the proposed design (network layer). AS, autonomous system; OSPF, open shortest path first; BGP, border gateway protocol.

### 2.3. Design for the QoS Support

The proposed design for the QoS support is as shown in Figure 3. To provide Internet services with the QoS support, the QoS class should be defined for them. The priority and characteristic for services should be then reflected in the scheduling and the resource allocation in the link and the network layers. For example, the resource for delay-sensitive services such as VoIP and video streaming services should be allocated by the CRA and the RBDC to minimize the delay [23,29,31]. Generally, the definition of the QoS class is different with each layer. The QoS mapping is thus needed between the network and link layers [1,23]. In the proposed design, the NMS manages the QoS class and the QoS policy such as the service priority, the maximum data rate, the QoS mapping, etc. This information is forwarded to the NCC and UTs. In the NCC and UTs for the QoS support, this information is used in the DAMA controller, the DAMA agent, and schedulers [1,23,29–31]. The data rate in the UT is limited by the policer in UTs to prevent the traffic congestion because the maximum data rate can be changed according to the type of terminal. For example, the maximum data rate can be changed because the maximum equivalent isotropically-radiated power (EIRP) is changed by the antenna size in the satellite network. In the log-on process for the QoS support, the NMS conducts the admission control of UTs by rejecting the log-on request message based on the total number of UTs and the sum of the maximum data rate for UTs. To design the QoS policy, the user requirement for each application is reflected by the QoS policy, as well as the goodput, and satisfaction for the user experience should be considered according to various environments for the application characteristics, the traffic pattern, and the protocol characteristics. Thus, in Section 3, we practically define the satisfaction ratio and the goodput for the user experience. We also measure the performance for various environments by varying each factor such as the application service type, the traffic load, the burstiness of the traffic, the number of users, and the resource allocation method in the link layer.

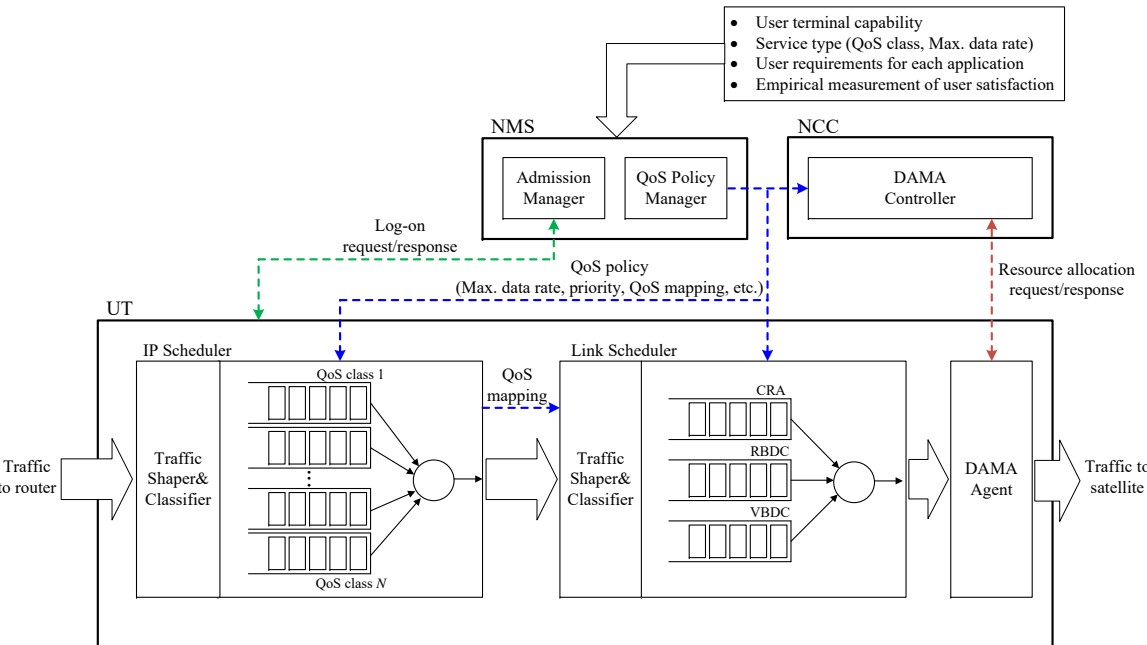

**Figure 3.** System model of the proposed design (QoS support). DAMA, demand assigned multiple access; UT, user terminals.

### 2.4. Other Considerations

In the satellite communication, to alleviate the effect of the long propagation delay and use the frequency resource efficiently, the usage of the performance enhancing proxy (PEP) should be considered in the transport and the application layers [21,36]. Generally, the TCP accelerator, the data compression, and the caching are used in the PEP in the satellite network [21,36]. Another concern

is the support of the OTM terminal in the satellite network [37,38]. In the OTM terminal, the active control system for the antenna pointing toward a satellite should be addressed [37]. Since temporary outage can occur by channel blockage due to pointing errors of the antenna and obstructions such as tunnels, high buildings, and trees, a data transfer mechanism such as an application layer forward error correction is also addressed in the satellite network to compensate the effect of temporary channel outage [38–40].

## 3. Performance Evaluation

In the performance evaluation, to show the QoS performance and derive the factors to be considered to the QoS policy in the satellite network, we analyzed the performance in the application layer for various environments and parameters in Table 1 [25,27,41–44]. We implemented an event-driven simulator in the Riverbed modeler Version 18.6. In the performance evaluation, source UTs send the data to destination UTs with the mesh connection using the MF-TDMA. We assumed that the total bandwidth and the control channel was 10 MHz and 3 MHz, respectively. The available bandwidth to transmit the service traffic can thus be calculated as:

$$B_A = (B_T - B_C) \times (1 - R_G) \times R_P, \tag{1}$$

where $B_T$ and $B_C$ mean the total bandwidth and the bandwidth to be allocated to transmit the control traffic, respectively. $R_G$ is the guard band ratio. $R_P$ is the MF-TDMA packing ratio. Thus, the available bandwidth is 6.044 MHz in the environment of Table 1. For example, if the spectral efficiency was assumed to be 1 bps/Hz, the maximum transmission capacity was 6.044 Mbps in the simulation. To evaluate a more realistic environment, we considered various simulation environments by varying each parameter such as the resource allocation method, the number of UTs, the number of service flows, the total traffic load, the data rate for each service flow, the burstiness, the spectral efficiency, and the delay constraint, as shown in Table 2. In this paper, the QoS requirement by users was the delay constraint. To evaluate the QoS performance, we measured the throughput, the 95th percentile end-to-end delay, the satisfaction ratio, and the goodput in the application layer. The throughput means the total throughput in the satellite network. The satisfaction ratio is defined as the ratio that all packets of transmitted data have successfully arrived at the destination UT while satisfying the delay constraint in the satellite network. The goodput means the total data volume that successfully has been received at the destination terminal within the delay constraint in the satellite network.

**Table 1.** The simulation parameters.

| Parameters | Values |
| --- | --- |
| Total bandwidth | 10 MHz |
| Guard band ratio | 10% |
| Control channel | 3 MHz |
| MF-TDMA packing ratio | 0.95 |
| Frame length | 0.5 s |
| Resource allocation delay | 4 frames |
| Propagation delay in the satellite link | 125 ms |
| Maximum data rate | 45 Mbps |
| The number of UTs | 50–150 |
| Spectral efficiency | 1, 2, 3 bps/Hz |
| The number of service flows | 50–150 |
| Ave.data TX rate for each service flow | 33–400 kbps |
| Ave. data size (application layer) | 2–500 KB |
| Ave. inter-arrival time | 0.5–10 s |

**Table 2.** The simulation scenario. VBDC, volume-based dynamic capacity; RBDC, a rate-based dynamic capacity.

| Parameters | Scen.1 | Scen. 2 | Scen. 3 |
|---|---|---|---|
| Resource allocation method | VBDC | RBDC | VBDC |
| Spectral efficiency (bps/Hz) | 1–3 | 1–3 | 1–3 |
| The number of UTs | 50 | 50 | 50–150 |
| The number of service flows | 50 | 50 | 50–150 |
| Total traffic load (Mbps) | 5–20 | 5–20 | 5–20 |
| Burstiness | 1–20 | 1–10 | 1 |
| Ave. data TX rate for each service flow (kbps) | 100–400 | 100–400 | 33–400 |
| Ave. data size (KB) | 6.25–500 | 6.25–250 | 2–25 |
| Ave. inter-arrival time (s) | 0.5–10 | 0.5–5 | 0.5–10 |
| Delay constraint (s) | 3–10 | 1–4 | 3–10 |

## 3.1. Performance Evaluation for the Delay-Tolerant Service

In Scenario 1, we show the performance of the delay-tolerant service according to the traffic load and the burstiness. The burstiness was from 1–20 for considering the traffic pattern of various applications. The VBDC was used in the resource allocation. The delay constraint was from 3–10 s with the consideration of the resource allocation delay. Figures 4 and 5 indicate the throughput and the 95th percentile end-to-end delay according to the traffic load for the spectral efficiencies of 1 bps/Hz, 2 bps/Hz, and 3 bps/Hz. In Figure 4, the burstiness does not highly affect the throughput. When the traffic load was more than the maximum transmission capacity, throughput values for the spectral efficiency of 1 bps/Hz, 2 bps/Hz, and 3 bps/Hz converged to about 5.5 Mbps, 12 Mbps, and 18 Mbps, respectively. However, the end-to-end delay was influenced by the burstiness, as shown in Figure 5. With increasing the burstiness, the 95th percentile end-to-end delay increased because the heavy traffic load that needs many frames for the transmission was instantaneously imposed on the satellite network in the case of high burstiness. When the traffic load was more than the maximum transmission capacity, the 95th percentile end-to-end delay exponentially increased. In this case, data were continually accumulated in the transmission buffer, resulting in the considerable queuing delay. Figures 6 and 7 show the satisfaction ratio and the goodput according to the traffic load for the spectral efficiency of 3 bps/Hz. Since the results for the spectral efficiencies of 1 bps/Hz, 2 bps/Hz, and 3 bps/Hz showed a similar tendency, the results for the spectral efficiency of 3 bps/Hz is only presented for the results of the satisfaction ratio and the goodput. It is shown that the satisfaction ratio and the goodput were reduced with increasing the burstiness due to the increase of the end-to-end delay by the burstiness. When the delay constraint decreased, these performances were degraded by the tight requirement, as shown in Figures 6 and 7.

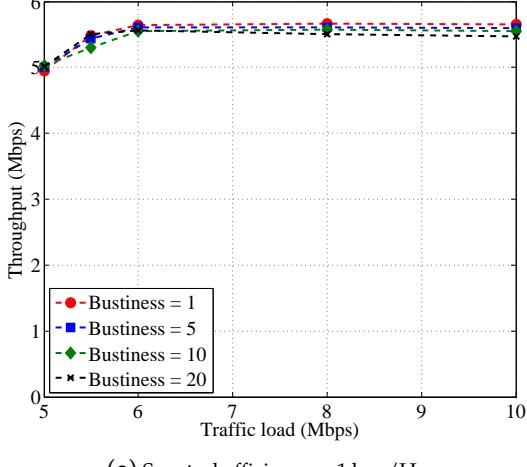

(**a**) Spectral efficiency = 1 bps/Hz

**Figure 4.** Cont.

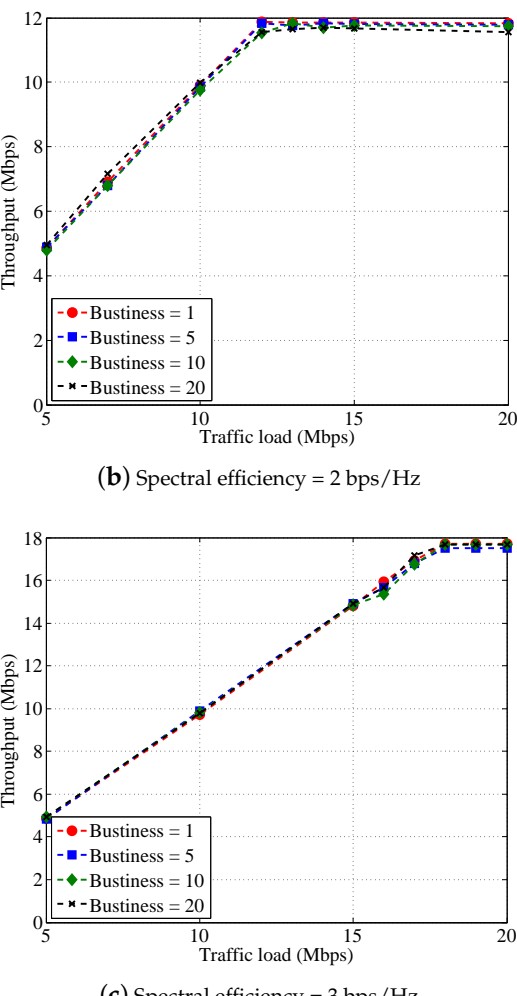

(**b**) Spectral efficiency = 2 bps/Hz

(**c**) Spectral efficiency = 3 bps/Hz

**Figure 4.** The network throughput at the application layer according to the traffic load (Mbps) when burstiness varies from 1–20 (VBDC).

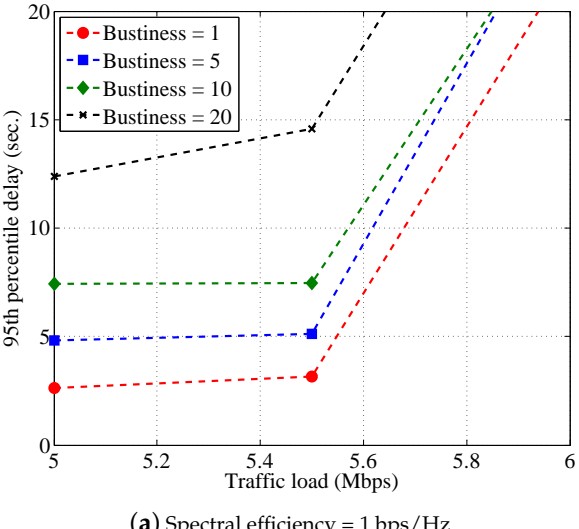

(**a**) Spectral efficiency = 1 bps/Hz

**Figure 5.** Cont.

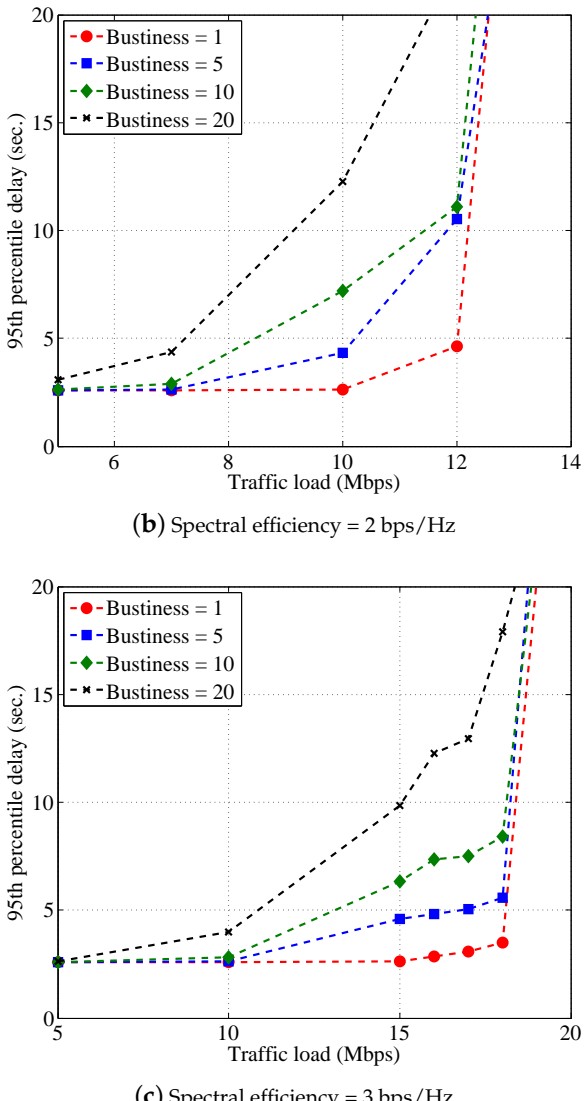

(**b**) Spectral efficiency = 2 bps/Hz

(**c**) Spectral efficiency = 3 bps/Hz

**Figure 5.** The 95th percentile delay according to the traffic load (Mbps) when burstiness varies from 1–20 (VBDC).

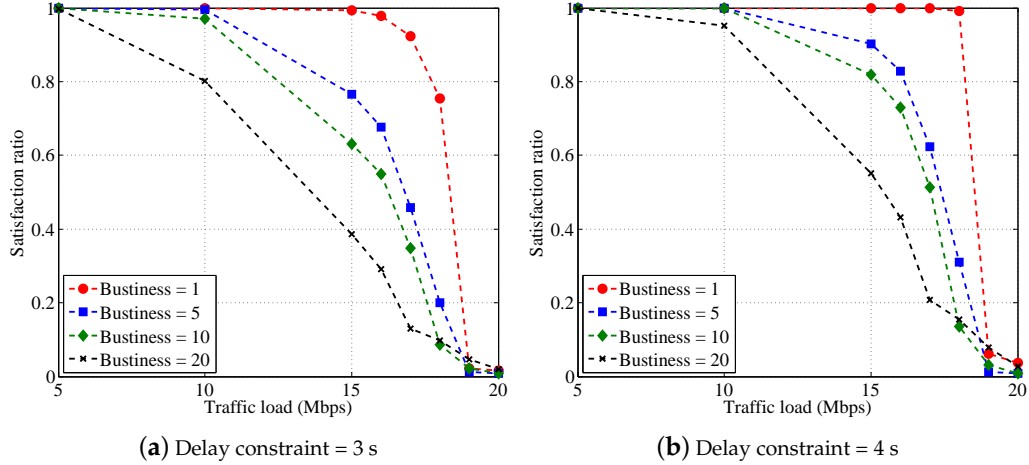

(**a**) Delay constraint = 3 s

(**b**) Delay constraint = 4 s

**Figure 6.** Cont.

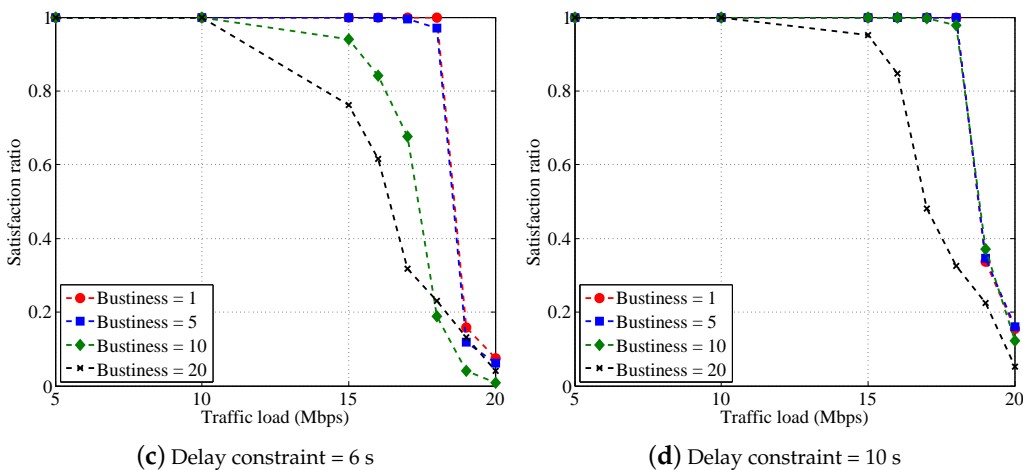

**Figure 6.** The satisfaction ratio when the spectral efficiency is 3 bps/Hz when the traffic load varies from 5–20 Mbps and burstiness varies from 1–20 (VBDC).

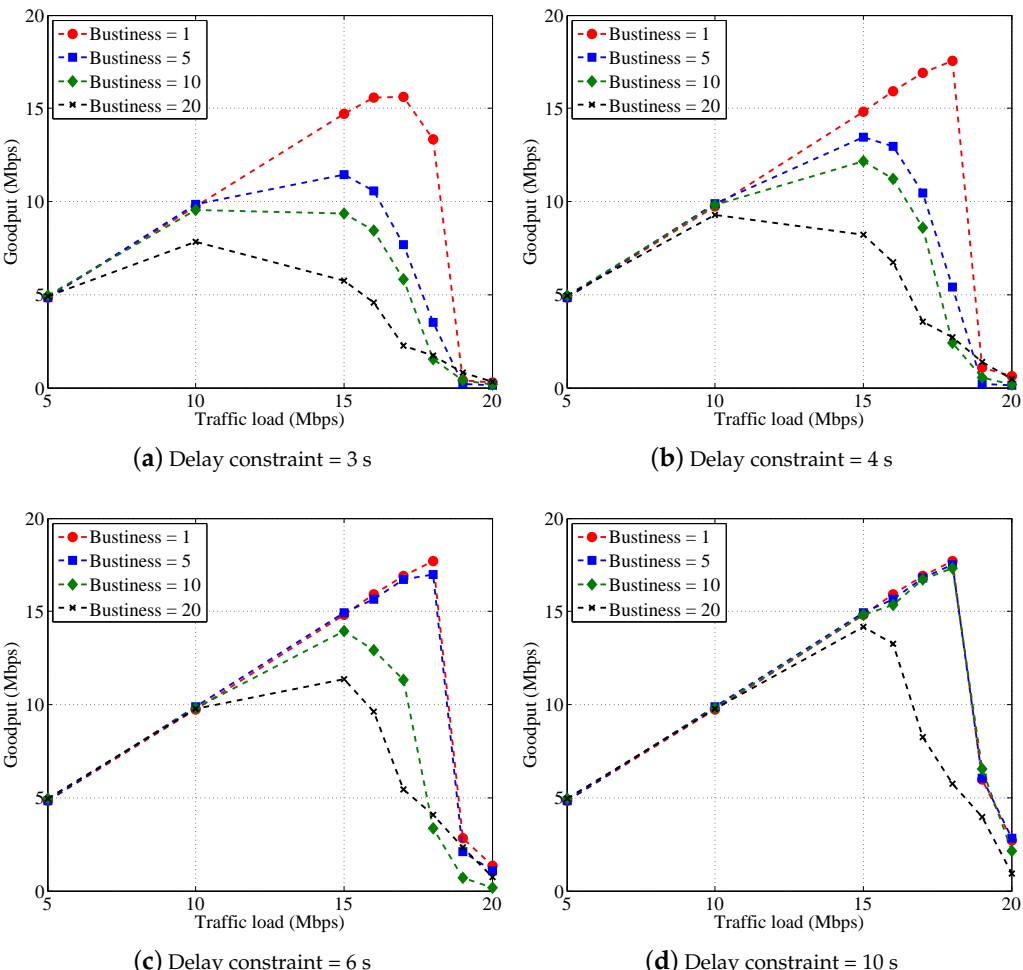

**Figure 7.** The goodput when the spectral efficiency is 3 bps/Hz when the traffic load varies from 5–20 Mbps and burstiness varies from 1–20 (VBDC).

Consequently, the factors such as the traffic load and the burstiness should be considered in the QoS policy for the delay-tolerant service in the satellite network. For example, to guarantee the QoS of UTs with the traffic with high burstiness, the number of UTs should be limited by the

admission control of UTs in the NMS considering the maximum goodput according to the burstiness. As mentioned above, since the heavy traffic load that needs many frames for the transmission is instantaneously imposed on the satellite network in the case of high burstiness, the end-to-end delay increased, resulting in the reduced maximum goodput.

### 3.2. Performance Evaluation for the Delay-Sensitive Service

In Scenario 2, we evaluated the performance of the delay-sensitive service allowing the long propagation delay in the satellite link according to the traffic load and the burstiness. The RBDC was applied in the resource allocation to eliminate the resource allocation delay. We selected a tight delay constraint that was from 1–4 s as compared to the Scenario 1 using VBDC. Since RBDC is generally applied to the resource allocation of the application such as VoIP and video streaming services, the burstiness was from 1–10, which is less than that of Scenario 1. In this paper, the average data rate was considered in the RBDC of the resource allocation for the efficient resource allocation. Figures 8 and 9 show the throughput and the 95th percentile end-to-end delay according to the traffic load for the spectral efficiencies of 1 bps/Hz, 2 bps/Hz, and 3 bps/Hz. In Figure 8, the burstiness does not highly affect the throughput. When the traffic load was more than the maximum transmission capacity, throughput values for the spectral efficiencies of 1 bps/Hz, 2 bps/Hz, and 3 bps/Hz converged like the results of Scenario 1. In Figure 9, it is indicated that the 95th percentile end-to-end delay is a little high. When the spectral efficiency, the burstiness, and the traffic load were 3 bps/Hz, 1, and 5 Mbps, respectively, the value was 1.073 s. In the case of the video streaming service, due to the data transmission of a variable bit rate (VBR), the heavy traffic load that needs many frames for the transmission can be temporarily imposed on UTs regardless of the burstiness, resulting in the increase of the end-to-end delay. For example, the data rate for the intra-frame transmission of the video streaming service was much more than the average data rate of the video service [45,46]. With similar trends as Scenario 1, the 95th percentile end-to-end delay increased with increasing the burstiness in this scenario. When the traffic load was more than the maximum transmission capacity, the 95th percentile end-to-end delay exponentially increased. Figures 10 and 11 show the satisfaction ratio and the goodput according to the traffic load for the spectral efficiency of 3 bps/Hz. It is shown that the maximum value of the satisfaction ratio and the goodput was reduced due to the imperfect estimation of the required resource when the delay constraint was tightly selected as 1 s [45,46]. The satisfaction ratio and the goodput were reduced with increasing the burstiness due to the increase of the end-to-end delay by the burstiness like Scenario 1. When the delay constraint increased, these performances improved with the loose requirement, as shown in Figures 10 and 11.

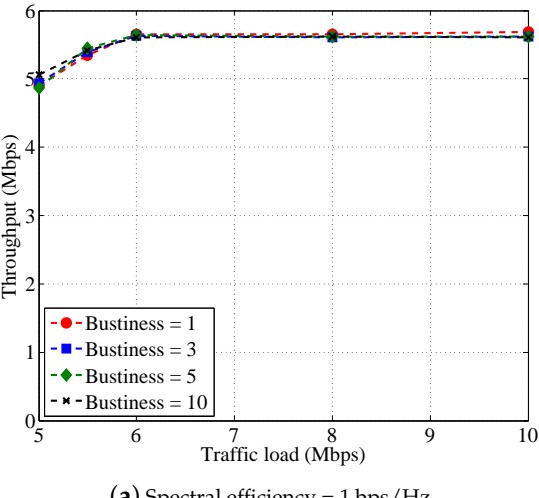

(**a**) Spectral efficiency = 1 bps/Hz

**Figure 8.** Cont.

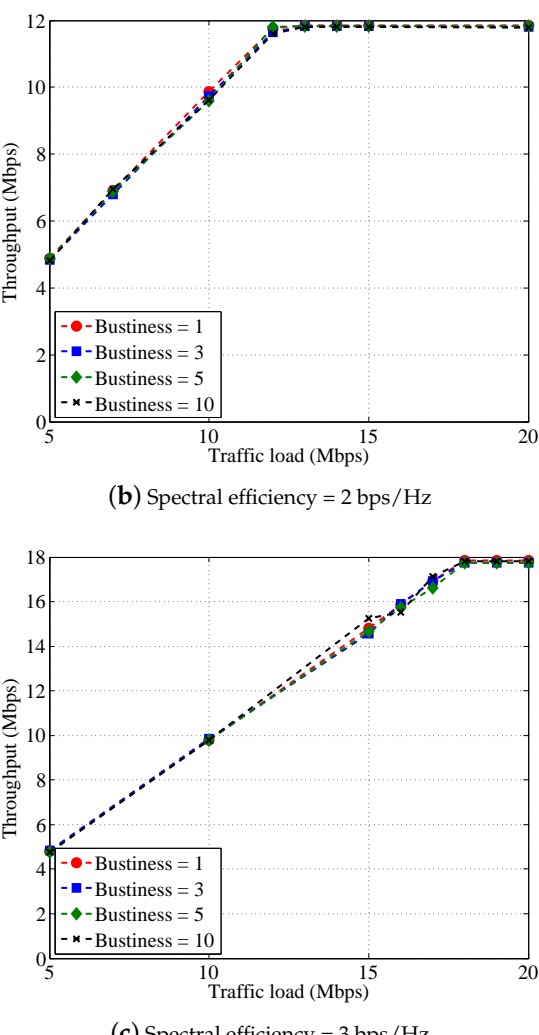

(**b**) Spectral efficiency = 2 bps/Hz

(**c**) Spectral efficiency = 3 bps/Hz

**Figure 8.** The network throughput at the application layer according to the traffic load (Mbps) when burstiness varies from 1–10 (RBDC).

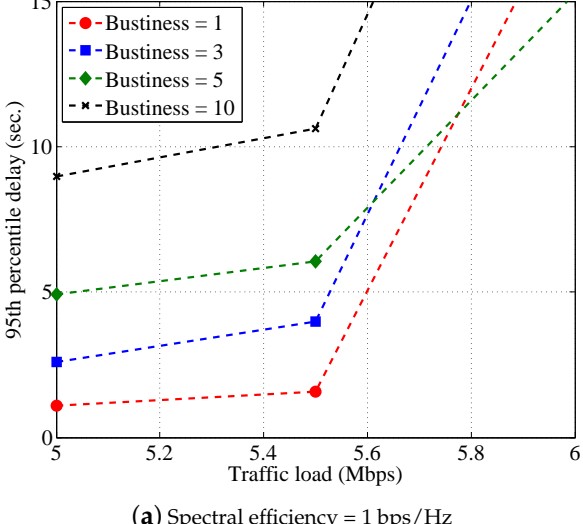

(**a**) Spectral efficiency = 1 bps/Hz

**Figure 9.** Cont.

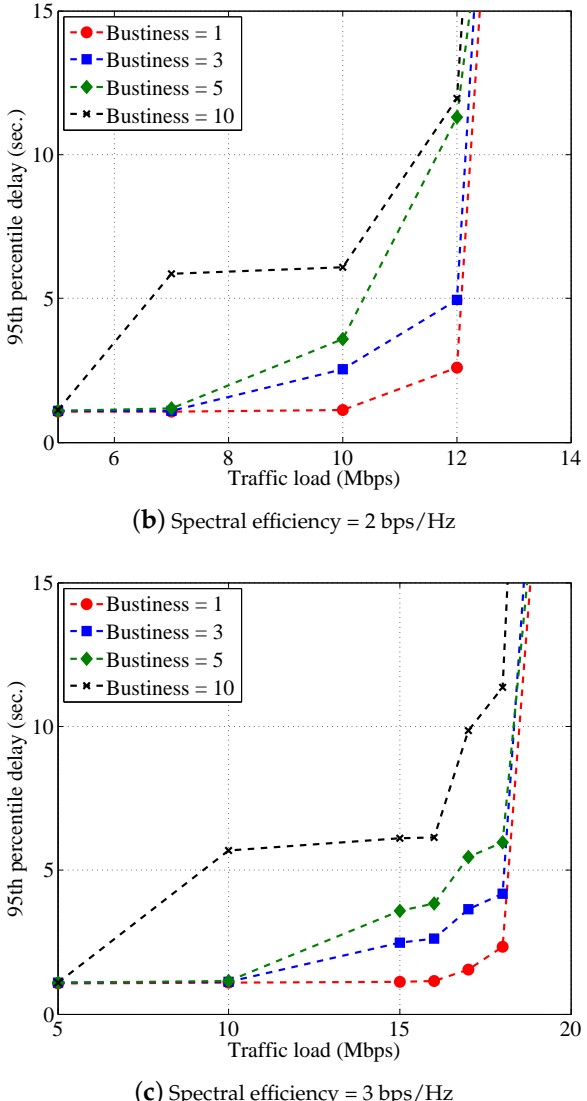

(**b**) Spectral efficiency = 2 bps/Hz

(**c**) Spectral efficiency = 3 bps/Hz

**Figure 9.** The 95th percentile delay according to the traffic load (Mbps) when burstiness varies from 1–10 (RBDC).

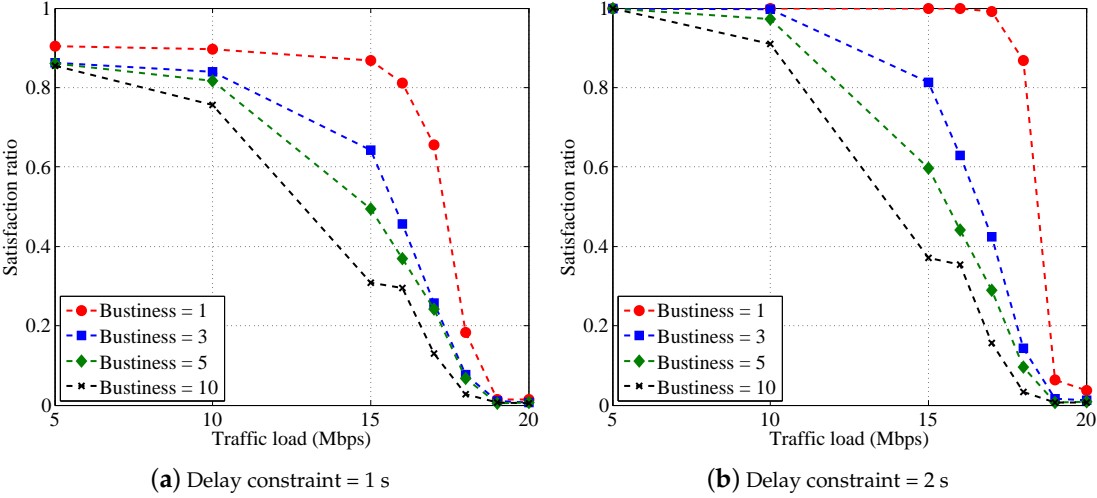

(**a**) Delay constraint = 1 s

(**b**) Delay constraint = 2 s

**Figure 10.** Cont.

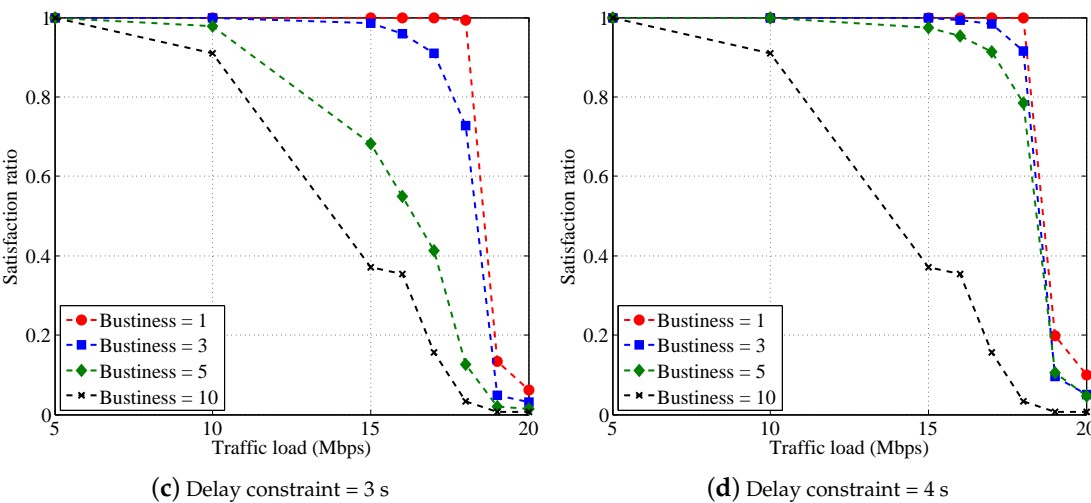

**Figure 10.** The satisfaction ratio when the spectral efficiency is 3 bps/Hz when the traffic load varies from 5–20 Mbps and burstiness varies from 1–10 (RBDC).

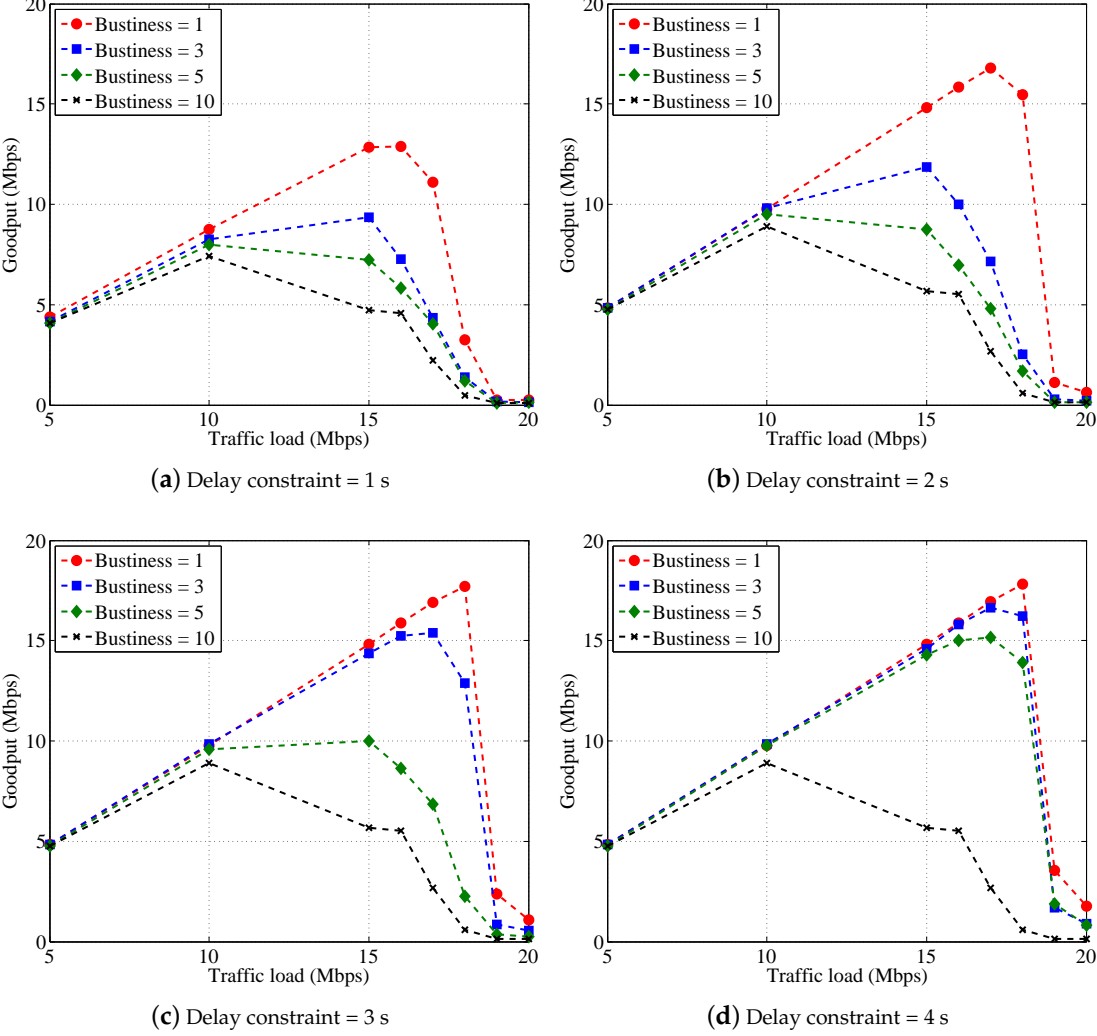

**Figure 11.** The goodput when the spectral efficiency is 3 bps/Hz when the traffic load varies from 5–20 Mbps and burstiness varies from 1–10 (RBDC).

As a result, the factor such as the estimation of effective data transmission rate should be additionally considered in the QoS policy for the delay-sensitive service, allowing the long propagation delay in the satellite network. For example, the RBDC based on average data rate is appropriate for the VoIP service. However, for the video streaming services, its VBR characteristic of the traffic generation and the burstiness of index frames should be considered in the estimation of the effective data transmission rate for RBDC. Thus, the QoS policy should select the suitable scheme to calculate the required resource according to the service.

### 3.3. Performance Evaluation According to the Number of UTs

In Scenario 3, we show the QoS performance according to the traffic load and the number of UTs in the satellite network. The VBDC was applied in the resource allocation. The burstiness was one. The delay constraint was from 3–10 s with consideration of the resource allocation delay. In this scenario, the total traffic load was constant and the data transmission rate of each UT was the total traffic load divided by the number of UTs. Thus, the data size to be transmitted in each ST decreased with increasing the number of UTs. Figures 12 and 13 show the throughput and the 95th percentile end-to-end delay according to the traffic load for the spectral efficiencies of 1, 2, and 3 bps/Hz. In Figure 12, it is shown that the maximum throughput was slightly reduced with increasing the number of UTs. For the spectral efficiency of 3 bps/Hz, the maximum throughputs for 50 UTs, 75 UTs, 100 UTs, 125 UTs, and 150 UTs were about 18 Mbps, 17.87 Mbps, 17.68 Mbps, 17.41 Mbps, and 17.13 Mbps, respectively. In the transmission of the data of a small size, the probability that the time slot of MF-TDMA was not fully filled increased, resulting in reducing the maximum throughput slightly. For this reason, the performance of the 95th percentile end-to-end delay in the heavy traffic was degraded by the number of UTs, as shown in Figure 13. Figures 14 and 15 show the satisfaction ratio and the goodput according to the traffic load for the spectral efficiency of 3 bps/Hz. As mentioned above, it was indicated that the performance of the satisfaction ratio and the goodput in the heavy traffic was degraded by increasing the number of UTs.

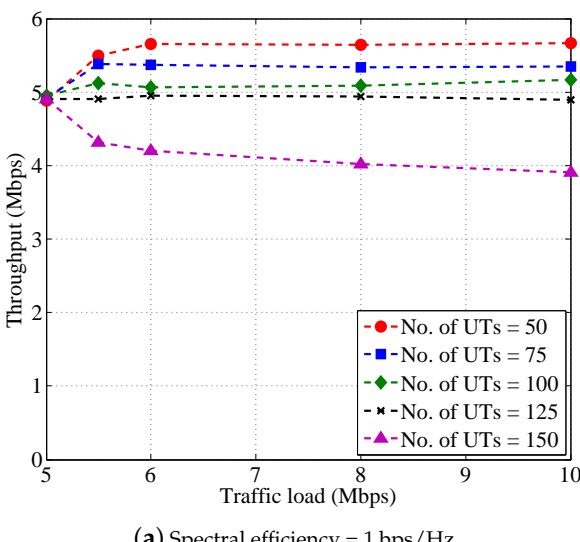

(**a**) Spectral efficiency = 1 bps/Hz

**Figure 12.** Cont.

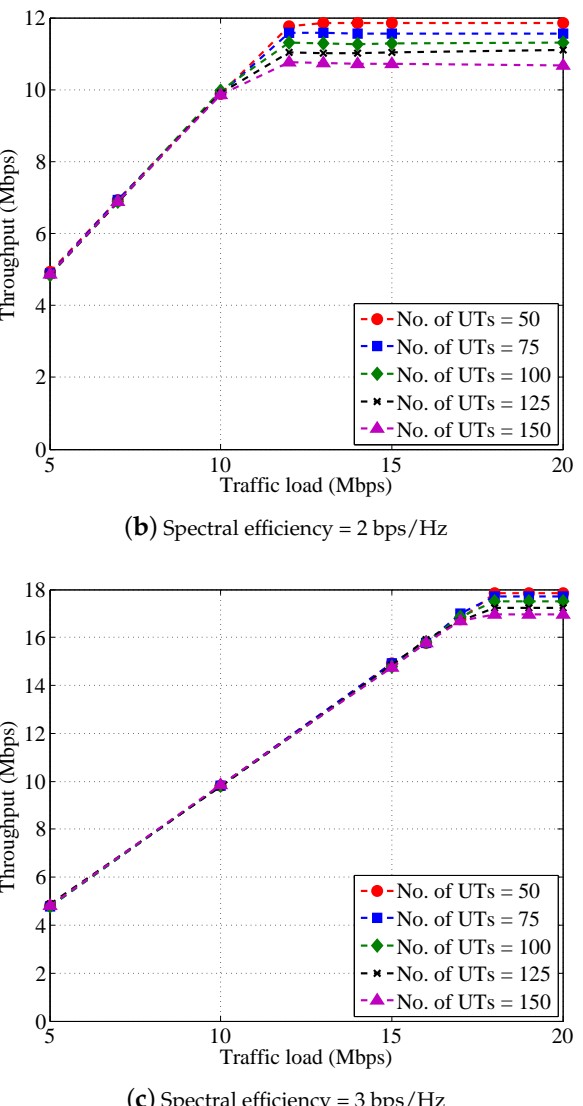

(**b**) Spectral efficiency = 2 bps/Hz

(**c**) Spectral efficiency = 3 bps/Hz

**Figure 12.** The network throughput at the application layer according to the traffic load (Mbps) when the number of UTs varies from 50–150 (VBDC).

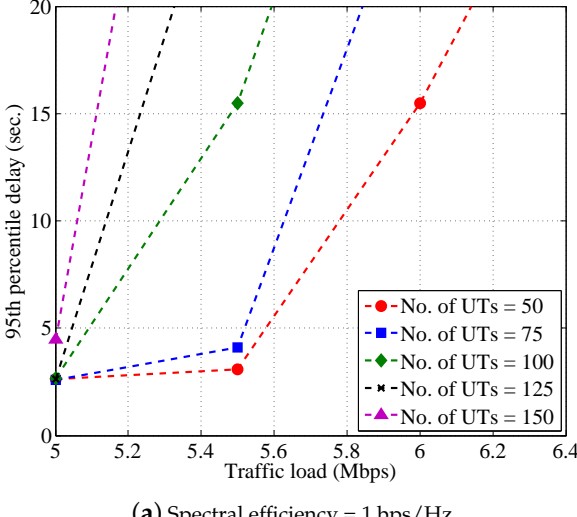

(**a**) Spectral efficiency = 1 bps/Hz

**Figure 13.** Cont.

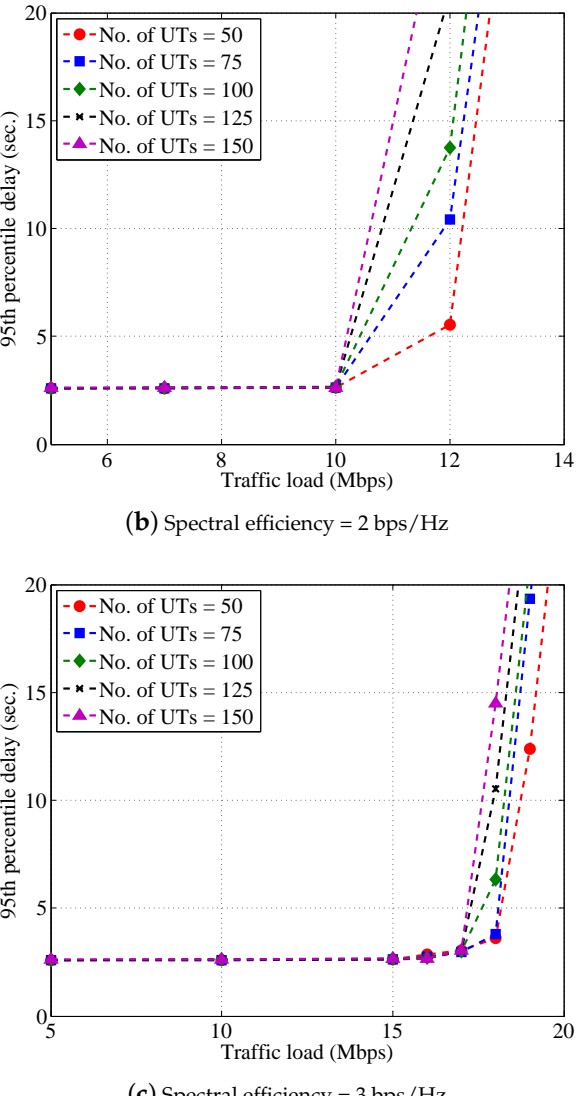

(**b**) Spectral efficiency = 2 bps/Hz

(**c**) Spectral efficiency = 3 bps/Hz

**Figure 13.** The 95th percentile delay according to the traffic load (Mbps) when the number of UTs varies from 50–150 (VBDC).

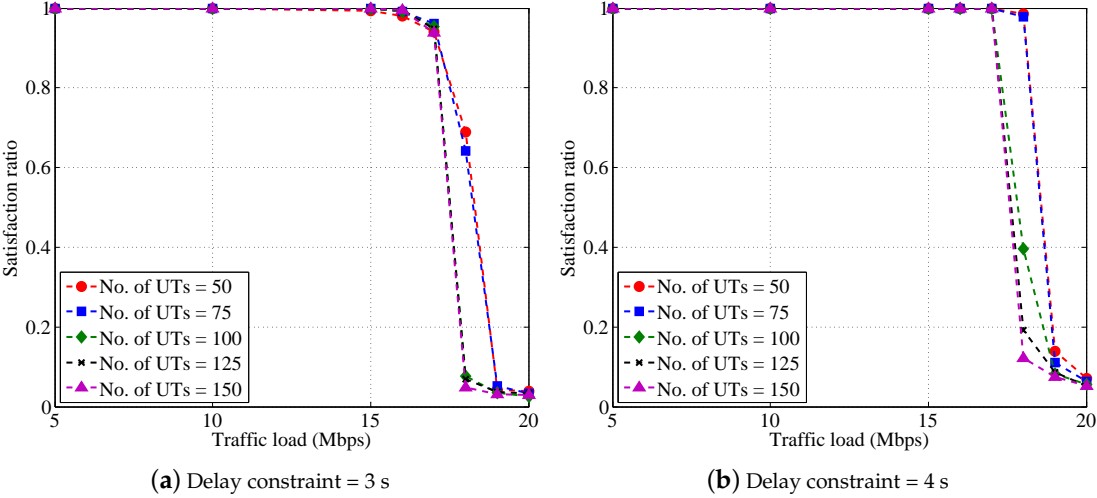

(**a**) Delay constraint = 3 s

(**b**) Delay constraint = 4 s

**Figure 14.** Cont.

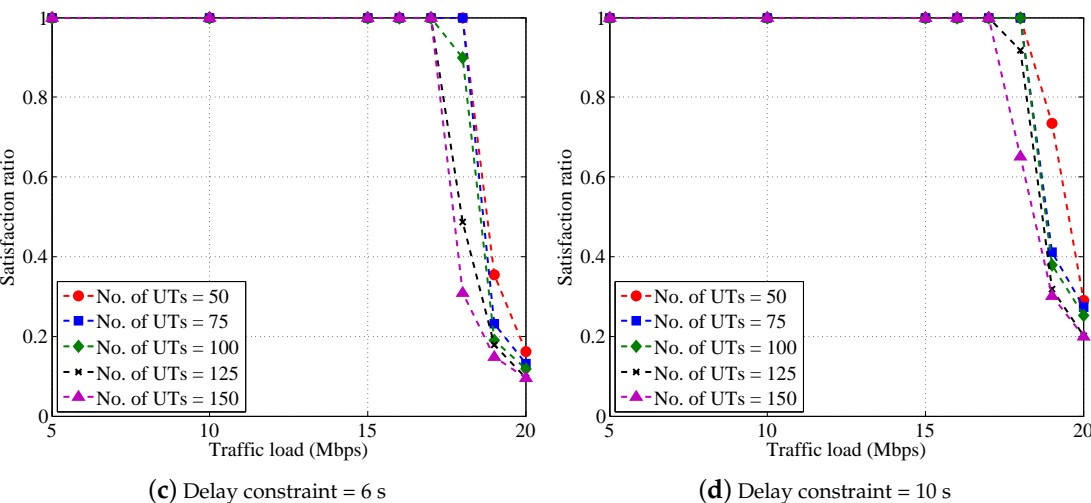

**Figure 14.** The satisfaction ratio when the spectral efficiency is 3 bps/Hz when the traffic load varies from 5–20 Mbps and the number of UTs varies from 50–150 (VBDC).

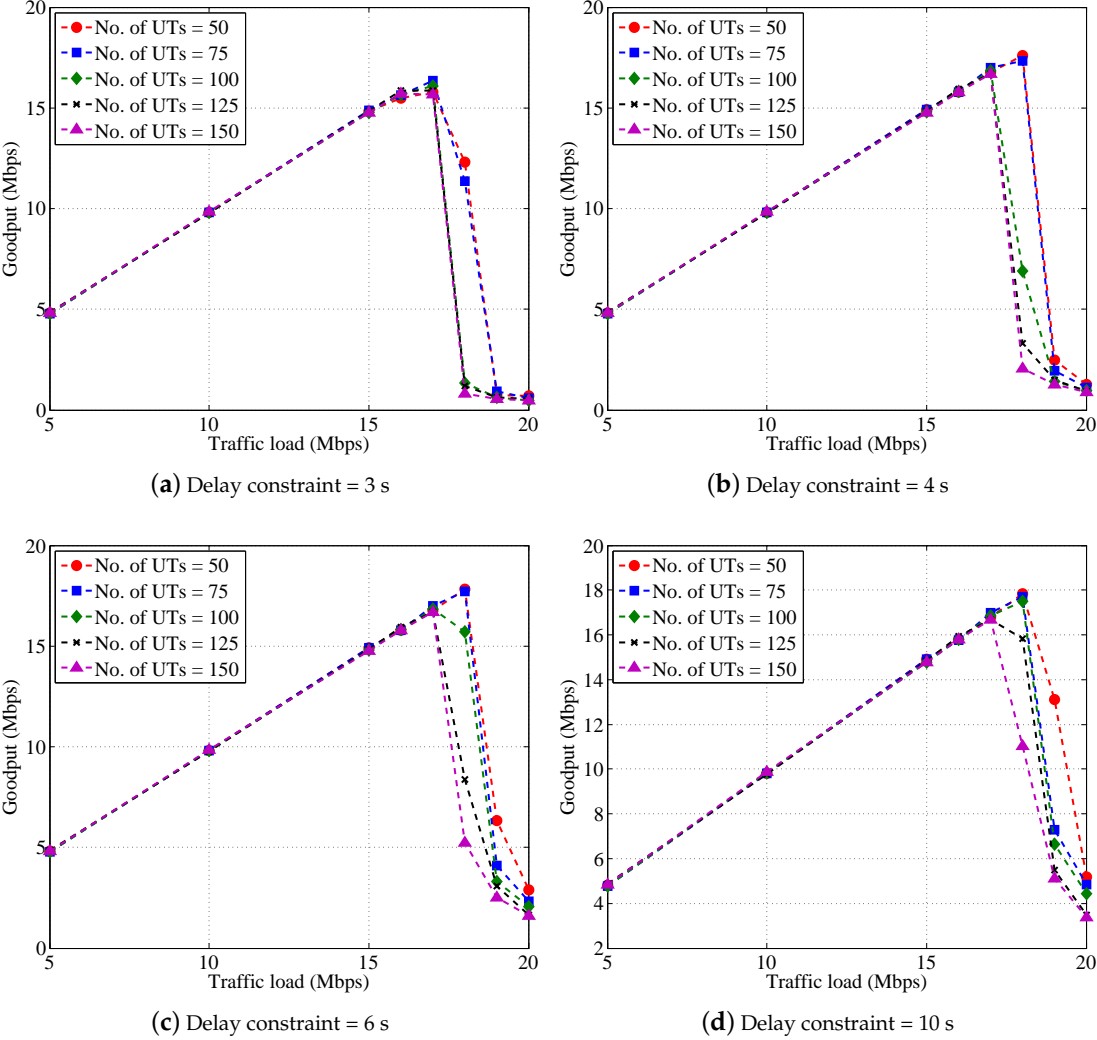

**Figure 15.** The goodput when the spectral efficiency is 3 bps/Hz when the traffic load varies from 5–20 Mbps and the number of UTs varies from 50–150 (VBDC).

Consequently, for IoT services such as remote control, telemetry, and surveillance of sensing data of a small size, the small data size had the effect of a slight reduction of the QoS performance. Its impact should be considered in the QoS policy in the satellite network. For example, by consideration of the data size and the maximum throughput in the network planing of the QoS policy, the number of UTs should be limited in the satellite network to prevent network congestion in dedicated networks for the IoT service.

## 4. Conclusions

In this paper, a satellite network to provide Internet services with the QoS support was addressed. In particular, we proposed a practical overall design of the MF-TDMA satellite network with the QoS support. To derive the major factor to be considered in the QoS policy, we defined the performance metrics for the user experience. In the various environments, by varying each factor such as the service type, the traffic load, the burstiness of the traffic, the number of users, and the resource allocation method in the link layer, the network performance for the user experience in the application layer was evaluated in the satellite network with the proposed overall design. Through the performance evaluation, it was shown that the factors such as the traffic load and the burstiness should be commonly considered in the QoS policy for delay-tolerant and delay-sensitive services in the satellite network. Results also showed that the estimation of the required resource to provide the QoS should be considered for the service using RBDC. It was shown that the small data size can reduce the QoS performance in the satellite network.

In satellite communication with its inherit characteristics such as the long propagation delay, it has not been easy to guarantee the QoS due to the resource allocation time, the large super-frame size, and the imperfect estimation of the required resource. Therefore, it is pointed out that the derived factors of this paper should be addressed in the QoS policy and the technique such as the admission control and estimation of required resource to enhance the QoS performance in the satellite network.

**Author Contributions:** K.-H.L. proposed the main idea, performed and evaluated the simulation experiments, and wrote the paper. K.Y.P. designed the validation methods and contributed to the discussion of this research.

**Funding:** This research received no external funding.

**Conflicts of Interest:** The authors declare no conflict of interest.

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
