# Peer review of "Overall Design of Satellite Networks for Internet Services with QoS Support"

_electronics, doi:10.3390/electronics8060683_

Round 1
Reviewer 1 Report
This paper studied the satellite network, which is able to provide Internet service for users in rural or other difficult-to-serve areas with guaranteeing a quality of service (QoS).
Authors proposed a practical overall MF-TDMA satellite network for Internet services with the QoS support. The proposed network was evaluated by practical and meaningful performance analysis considering control and protocol overheads. Besides, users' satisfaction for various environments in the application layer was also evaluated.
Overall, the paper is clearly stated and the design of the satellite network is interesting. The performance of the proposed satellite network was also evaluated appropriately.
There are several suggestions.
1. Page 1, line 34: "substantial" should be "Substantial".
2. What is the main novelty of this paper compared with previous work in the literature? Can authors provide one short paragraph to summarize it?
3. Is it possible to conduct comparison experiments by considering the existing satellite networks?
I suggest accepting the paper after a major revision.
Author Response
Authors’ Reply to Editor and Reviewers’ Comments
Most of all, we would like to thank the associate editor and the reviewers to spend their valuable time and effort to review our paper. We have carefully read all of reviewers' comments and have revised our manuscript according to reviewers' insightful comments and suggestions. Please find below our detailed replies to each of the comments. Once again, we appreciate your kind and careful suggestions.
Note that we use independent reference numbers for this reply letter with respect to the original submission. Also, we denote tables and figures of this reply letter by terms of “Table 1” and “Figure 1”, and we denote the tables and figures of the paper by terms of “TABLE I” and “Fig.1”.
Detail revisions are following.
Reviewer #1 comment #1: Page 1, line 34: “substantial” should be Substantial” |
Reply 1-1:
We appreciate for your comments. Following the reviewer comment, we have fixed typo.
Revision 1-1:
Before | After |
i.[p. 1, line 34] substantial researches have focused on needed ~ | i.[p. 1, line 34] Substantial researches have focused on needed ~ |
Reviewer #1 comment #2: What is the main novelty of this paper compared with previous work in the literature? Can authors provide one short paragraph to summarize? |
Reply 1-2:
We really appreciate for your valuable comments. In our paper, to provide Internet services with the QoS support via satellite networks, we would like to estimate the network level performance for satellite network which consists of the end to end network architecture. Thus, we had modelized our system with considerations for L1~L7 issues related to QoS. Furthermore, we also would like to evaluate the user satisfaction in the application layer in the paper. TABLE 1 shows the novelty of the paper as compared with the previous works. However, in the previous version of the paper, the description of novelty is not clear. Thus, we have included additional explanations in the Introduction (Section 0) more clearly.
Table 1. Comparisons between our paper and previous works.
Focused research areas for QoS | Performance evaluation and design for QoS | |||
Whole network-level performance evaluation and design | Evaluation in the application layer. | Evaluation for the user satisfaction | ||
the proposed paper | L1: spectral efficiency/ channel(control, data) architecture L2: resource allocation(log-on, DAMA, MF-TDMA) L3: routing architecture L4: performance enhancing proxy, on-the-move terminal L7: traffic modeling (burstiness, traffic load) | O | O | O |
[13] | L1:Code design | X | X | X |
[14] | L1: Power control | X | X | X |
[15] | L2: Gateway handover | X | O | X |
[16] | L2: Admission control | X | X | X |
[17],[18] | L3: Routing | X | X | X |
[19],[20] | L4: TCP/ congestion control | X | O | X |
[20] | L4/L7: Caching | X | O | O |
[23],[24] | L7: VoIP | X | O | X |
Revision 1-2:
Before | After |
i.[p. 2] None. | i.[p. 2, line 63~67] To provide Internet services with the QoS support via satellite networks, it is needed to estimate the network level performance for satellite network which consists of the end to end network architecture. However, to the best of our knowledge, it has not been explored that whole network-level performance evaluation and designs for the QoS in the satellite network according to various applications of Internet services. |
Reviewer #1 comment #3: Is it possible to conduct comparison experiments by considering the existing satellite network? |
Reply 1-3:
We really appreciate for your meaningful comments. Actually, authors tried to perform a network-level performance comparison with the actual satellite network to verify our modeling as you asked. As you know, in commercial satellite networks, performance at the network level is set to confidential for each company, making performance comparison difficult. However, in network modeling, each element, such as satellite terminal, modem, and so on, are modeled as they are used in Korea, and accuracy of actual satellite network performance is improved.
Revision 1-3:
Before | After |
i.[p. 1, line 34]
| i.[p. 1, line 34]
|
Reviewer 2 Report
The authors propose a design of the satellite network to provide Internet services with via satellite network. They consider two service types: delay-tolerant and delay-sensitive services.
The English should be improved. For example, “delay-tolerance” change into delay-tolerant.
There are typos in the manuscript.
“addressed. substantial researches have focused on needed”
The authors discuss proving two types of services with different QoS. The recent concept of network slicing [1] is invented in order to meet the QoS of different services over the same infrastructure. The authors should refer to recent papers for network slicing.
[1] D. Gligoroski and K. Kralevska, “Expanded combinatorial designs as tool
to model network slicing in 5g,” IEEE Access, vol. 7, pp. 54 879–54 887,
2019.
How does this manuscript relate to other works by the same authors? What is the difference?
Author Response
Authors’ Reply to Editor and Reviewers’ Comments
Most of all, we would like to thank the associate editor and the reviewers to spend their valuable time and effort to review our paper. We have carefully read all of reviewers' comments and have revised our manuscript according to reviewers' insightful comments and suggestions. Please find below our detailed replies to each of the comments. Once again, we appreciate your kind and careful suggestions.
Note that we use independent reference numbers for this reply letter with respect to the original submission. Also, we denote tables and figures of this reply letter by terms of “Table 1” and “Figure 1”, and we denote the tables and figures of the paper by terms of “TABLE I” and “Fig.1”.
Detail revisions are following.
Reviewer #2 comment #1: The English should be improved. For example, “delay-tolerance” change into delay-tolerant There are typos in the manuscript. “addressed. Substantial researches have focused on needed” |
Reply 2-1:
We appreciate for your comments. Following the reviewer comment, we have fixed typos and improved sentences.
Revision 2-1:
Before | After |
i.[p. 1, line 34] substantial researches have focused on needed ~ | i.[p. 1, line 34] Substantial researches have focused on needed ~ |
ii.[overall paper] delay-tolerance | ii.[overall paper] delay-tolerant |
Reviewer #2 comment #2: The authors discuss proving two types of services with different QoS. The recent concept of network slicing[1] is invented in order to meet the QoS of different service over the same infrastructure. The authors should refer to recent papers for network slicing. [1] D. Gligoroski and K. Kralevska, “Expanded combinatorial designs as tool to model network slicing in 5g ”, IEEE Access, vol.7, pp 54 879-54- 887, 2019 |
Reply 2-2:
We really appreciate for your valuable comments. Following the reviewer comment, we have included additional reference paper for the network slicing for the QoS.
Revision 2-2:
Before | After |
i.[p. 1, line 18-19] In the future network, it will be necessary to offer tremendously more capacity than current ones to satisfy increasing traffic demanded by users and new applications [1–2]. | i.[p. 1, line 18-20] In the future network, it will be necessary to offer tremendously more capacity than current ones to satisfy increasing traffic demanded by users and new applications with a quality of service (QoS) [1–4]. |
[Reference] None. | [Reference] [3] Foukas, X.; Patounas, G.; Elmokashfi, A.; Marina, M.K. Network Slicing in 5G: Survey and Challenges. IEEE Communications Magazine 2017, 55, 94–100. doi:10.1109/MCOM.2017.1600951. [4] Gligoroski, D.; Kralevska, K. Expanded Combinatorial Designs as Tool to Model Network Slicing in 5G. IEEE Access 2019, 7, 54879–54887. doi:10.1109/ACCESS.2019.2913185. |
Reviewer #2 comment #3: How does this manuscript relate to other works by the same authors? What is the difference? |
Reply 2-3:
In our previous works [R1]-[R3], we focus on the applicability of the application layer forward correction (AL-FEC) to enhance the reliability of the data transfer in the satellite communication system with the link blockage environment. It can cause packet loss in the satellite link, resulting in depredating the QoS performance. Thus, this issue is related to the design of satellite networks for Internet services with QoS support. In the previous works, the protocol for the reliability of the data transfer is only focused. Overall network architecture and performance for QoS do not considered in the previous works. That is the distinguished difference between this manuscript and previous works.
References of Reply Letter
[R1] Lee, K.H.; Kim, J.M.; Kim, J.H. Transfer time analysis of file transfer framework with AL-FEC in SOTM networks. EURASIP Journal on Wireless Communications and Networking 2015, 2015, 1–8. doi:10.1186/s13638-015-0462-7.
[R2] Lee, K.; Kim, J. Efficient AL-FEC Mechanism Aided by Navigation Systems for SOTM Systems. IEEE Transactions on Wireless Communications 2016, 15, 6651–6661. doi:10.1109/TWC.2016.2586845.
[R3] Lee, K.H.; Jang, D.H.; Lee, S.J.; Cha, J.R. Target BER selection scheme in LMS networks using AL-FEC systems. Computer Networks 2017, 127, 190–199. doi:https://doi.org/10.1016/j.comnet.2017.08.005.
Round 2
Reviewer 1 Report
Authors have addressed my concerns proposed in the previous round review. The current version looks fine. I suggest accepting the paper in the current form.
Reviewer 2 Report
The reviewer is satisfied with the current version of the manuscript.